# The Effect of Cooling Rate on the Microstructure Evolution and Mechanical Properties of Ti-Microalloyed Steel Plates

**DOI:** 10.3390/ma15041385

**Published:** 2022-02-13

**Authors:** Xiaolin Li, Qian Li, Haozhe Li, Xiangyu Gao, Xiangtao Deng, Zhaodong Wang

**Affiliations:** 1Center of Advanced Lubrication and Seal Materials, State Key Laboratory of Solidification Processing, Northwestern Polytechnical University, Xi’an 710072, China; liqian0410@mail.nwpu.edu.cn (Q.L.); lihaozhe@mail.nwpu.edu.cn (H.L.); 2Analytical & Testing Center, Northwestern Polytechnical University, Xi’an 710072, China; tsgxgaoxiangyu@nwpu.edu.cn; 3State Key Laboratory of Rolling and Automation, Northeastern University, Shenyang 110819, China; zhdwang@mail.neu.edu.cn

**Keywords:** thermomechanical controlled processing (TMCP), ultra-fast cooling system, microstructure evolution, precipitation behavior, mechanical properties

## Abstract

Ti-bearing microalloyed steel plates with a thickness of 40 mm were subjected to ultra-fast cooling (UFC) and traditional accelerate cooling after hot-rolling, aiming to investigate the effect of cooling rate on the microstructure and mechanical properties homogeneity, and thus obtain thick plates with superior and homogeneous mechanical properties. Yield strength, tensile strength, and elongation were 642 MPa, 740 MPa, 19.2% and 592 MPa, 720 MPa and 16.7%, respectively, in the surface and mid-thickness of the steel with ultra-fast cooling, while in the steel with traditional accelerate cooling, 535 MPa, 645 MPa, 23.4% and 485 MPa, 608 MPa, 16.2% were obtained in the surface and mid-thickness of the plate. The yield strength has been greatly improved after UFC, for the refinement of grain and precipitates produced by UFC. In addition, the equivalent grain size and precipitates size in the thick plate with UFC are homogeneous in the thickness direction, leading to uniform mechanical properties. The crystallographic characteristics of different precipitates have been studied. The precipitates formed in the austenite deformation stage obey Kurdjumov–Sachs orientation relationship with the ferrite matrix, while the fine precipitates formed in the ferrite obey [112]_MC_//[110]_α_ and ${\left(\bar{1}\bar{1}1\right)}_{\mathrm{MC}}$//${\left(\bar{1}12\right)}_{\alpha }$ orientation relationship with the ferrite matrix.

## 1. Introduction

During recent decades, the adjustment and upgrade of steel products have always been essential tasks in the iron and steel industry under the pressure with respect to resources, energy, and environment. Therefore, it is essential to propose an effective way to strengthen the microalloyed steel and make the microalloyed steel develop continuously and positively. Funakawa in JFE Corporation developed an automobile steel with a yield strength of 780 MPa. The steel was strengthened by the addition of Ti and Mo elements, and the microstructure of the steel was ferrite with high density randomly distributed nano-precipitates [1]. Besides microalloying, the thermomechanical controlled processing (TCMP) technique has also been proposed and studied extensively. Wang and his group at Northeastern University studied the effect of TMCP parameters, including hot-rolling temperature [2,3], cooling rate [3], and finish cooling temperature [4,5,6], on the mechanical properties of medium thickness microalloyed steel plates. Nishioka published a review describing the metallurgical aspects of the microalloying in steel, and discussing the advantages of TMCP, for example, in terms of weldability, which is reduced upon alloying [7]. Even though extensive studies have been performed on the medium thickness plate, the research on the thick steel plate is still limited.

Advanced thick steel plates are important structural materials for ship hulls, pressure vessels, bridges, buildings, and offshore structures [8,9,10,11]. The general trend for them is achieving higher strength and maintaining or improving other properties simultaneously, and also guaranteeing homogeneity in thickness. TMCP technology is a simple and cost-effective method to enhance the properties of steel [12,13,14], which was first reported in Japan. Uemori et al. [15] in Nippon Steel Co., reported that the yield strength of microalloyed heavy steel plates can increase by 300 MPa compared with the traditional quenching-tempering steel. While in TMCP of advanced heavy steel plates, the deformation and cooling are rarely uniform over the entire thickness, which then leads to inhomogeneous microstructure and mechanical properties being obtained in the plate. Thus, In order to enhance the homogeneity of the heavy steel plate, expensive elements, such as Ni, Cr, Mo, and Cu were added to improve the hardenability of the steel plate [16,17,18]. Although a satisfactory strength-toughness combination is obtained in the steel plate arising from the combined effect of refined ferrite grain size and precipitation strengthening, the weldability is still a tough problem that needs to be overcome due to the enhanced carbon equivalent by the addition of expensive microalloying elements. So, the technique should be developed to increase the homogeneity of the thick steel plate instead of adding too many hardenability elements.

Based on the above problem, new generation TMCP (NG-TMCP), that is TMCP involving ultrafast cooling (UFC) technology, is currently being applied to industrial production [19,20,21], aiming to reduce the consumption of expensive alloying elements, improve the homogeneity of the steel plate, and make the steel making process economically visible [15,22,23,24]. The cooling medium for UFC is water, and the finish cooling temperature was controlled by changing the pressure and density of the water outlet, which was determined by the automatic control system. The principle of the UFC device is to reduce the aperture of the water outlet, densify the water outlet and increase the water pressure, to ensure that the water flow has enough energy and impact force to break the water film. In addition, NG-TCMP can greatly suppress the degree of the alloys precipitated in austenite and increase the supersaturating in ferrite or bainite, improving the precipitation hardening. Thus, it is of practical significance to evaluate the effect of UFC after hot-rolling on the microstructural and mechanical properties homogeneity in heavy steel plates.

In the present work, we focus on comparing the effect of UFC and traditional accelerated cooling on the homogeneity of microstructure and mechanical properties of thick plates, and investigate the correlation between microstructure and mechanical properties. Furthermore, the precipitation behavior in different regions of the thick plate is also studied, with a particular focus on the crystallographic characteristic.

## 2. Materials and Methods

### 2.1. Materials and Thermo-Mechanical Processing

The chemical composition of the investigating steel is listed in Table 1. 0.08% Ti was added to the composition of traditional Q345 grade steel. The steel was melted, cast, and then rolled into plates with NG-TMCP and traditional TMCP technologies. The detailed processing is as follows: the steel ingot was isothermally held at 1250 °C for 2 h to dissolve the inclusions formed in the casting stage, and then hot-rolled to 40 mm with a thickness reduction of 60%. Next, the steels were cooled to ~590 °C with ultra-fast cooling and traditional cooling, and held for 20 min (hereafter referred to steel A and B). Finally, the steels were air-cooled to room temperature. The schematic diagram in Figure 1 illustrates the difference between NG-TMCP and traditional TMCP, and the detailed parameters of the processing are given in Table 2.

### 2.2. Microstructural Characterization

The specimens for microstructure characterization were cut from the hot-rolled plates, and then mechanically ground with sandpapers possessing different roughness in order to meet the surface quality before etching. The optical micrograph (OM; Olympus, Tokyo, Japan) and scanning electronic micrograph (SEM) specimens were etched with 4% nital solution by volumeat room temperature, and conducted by Zeiss Ultra 55 SEM (Oberkochen, Baden-Württemberg, Germany). For electron backscattered diffraction (EBSD; Oberkochen, Baden-Württemberg, Germany) detections, the specimens were electrochemically polished in a solution consisting of 650 mL ethyl alcohol, 100 mL perchloric acid, and 50 mL distilled water. TEM characterizations were carried out by FEI Tecnai G^2^ F20 TEM (Hillsboro, OR, USA) equipped with an energy-dispersive X-ray (EDX) spectrometer at an accelerating voltage of 200 kV. The specimens were cut from the steel plates and mechanically thinned to ~0.06 mm. Then the foils were electrochemically jet polished at −30 °C in a solution containing 9 vol.% Perchloric acid in ethanol.

The TEM technique was used to investigate the category, crystal structure, and lattice parameter of the precipitates. A detailed description of the method has been listed in previously published papers [4]. In addition, Crystal Maker software was used for the simulation of the unit cell structure to clearly present the orientation relationship between the precipitates and ferrite matrix.

### 2.3. Hardness and Tensile Tests

The hardness was tested by a Vickers hardness tester with a working load of 1 N and a dwelling time of 10 s, and each sample was tested at least five times. Cylindrical tensile test specimens with a diameter of 5 mm and a gauge length of 25 mm were prepared from the surface and mid-thickness of the thick steel plates perpendicular to the rolling direction and the specimens are parallel with the plate. The tensile test was conducted in SANS-5000 (Zhejiang, China) tensile tester with a rate of 1 mm/min. The impact toughness test was conducted in Instron 9250 impact tester (Norwood, MA, USA), and the Standard Charpy v-notch impact sample was chosen with a dimension of 10 × 10 × 55 mm^3^, which were prepared along the rolling direction to determine impact toughness at −20 and −40 °C.

## 3. Results and Discussion

### 3.1. Microstructural Evolution

Figure 2 presents the microstructure of the specimens obtained from the surface and mid-thickness of steel A. The microstructure in the surface region primarily consists of lath bainite (LB) and polygonal ferrite (PF), together with a little dark-etched pearlite (P), as shown in Figure 2a,b. TEM micrograph in Figure 2c indicates upper bainite structure for some thin cementite located within ferrite laths, indexed by the selected area electronic diffraction (SAED) pattern in Figure 2c. Figure 2d–f illustrates the microstructure in the mid-thickness region. The microstructure consists of PF, P, acicular ferrite (AF), and the corresponding percentage were ~89%, ~8%, and ~3%, respectively. Figure 2f shows the TEM image of PF and P. High-density dislocations located in PF and pearlitic ferrite. The lamellar cementite locates in ferrite laths in pearlite and the interlamellar spacing is ~200 nm.

Figure 3 shows the microstructure of the specimens obtained from the surface and mid-thickness region of steel B. The microstructure in the surface region consists of AF and PF, with a small amount of dark-etched P (Figure 3a). EDSD results indicate that there is no obvious texture in the microstructure, and the average grain size is 13.2 μm (Figure 3b). Figure 3c shows the corresponding TEM image, PF and AF lath can be observed, accompanied by extensive dislocations in AF lath. In order to clearly observe the dislocations, a two-beam diffraction method with the *g* = (002) has been used, as shown in Figure 3d. Figure 3e–h shows the microstructure in the mid-thickness region. The microstructure mainly consists of PF and P. The average size of the grain is 18.6 μm, as shown in Figure 3e, f. Figure 3g, h present the TEM images of PF. Extensive dislocation has been observed with *g* = (110). The grain size in the mid-thickness region is larger compared with that in the surface of the plate, which reveals obvious inhomogeneity in the microstructure of steel B. The inhomogeneity in microstructure can be attributed to different finish cooling temperatures and cooling rates. In the mid-thickness of the plate, no AF has been observed due to the lower undercooling, and a large amount of pearlite formed by the relatively lower cooling rate.

### 3.2. Mechanical Properties

Figure 4 presents the hardness distribution along the thickness direction in steel A and B. In steel A, the hardness at surface, 1/8, 1/4, 3/8 and 1/2 thickness is 267, 256, 246, 239, and 235 HV, respectively. While in steel B, the hardness is 229, 211, 198, 186, 179 HV, respectively. The hardness decreases with the increase of distance to the surface in both steels, which means that the hardness in the surface region is higher compared with that in the mid-thickness region. The hardness difference for the surface and mid-thickness region is 38 HV and 50 HV in steels A and B, respectively. It is visible that the hardness is more homogeneous in steel A, compared with that in steel B.

The mechanical properties of steel A and B are listed in Table 3, with the data obtained from three specimens. The tensile stress–strain curves of specimens in the surface and mid-thickness regions of steel A and B are presented in Figure 5. In Figure 5a, obvious yield platform in tensile strain–stress curve has observed both in the surface and mid-thickness region of steel A, while continuous yielding behavior observed in steel B (Figure 5b). In steel A, the yield strength, tensile strength, and elongation for the surface and mid-thickness region are 642 MPa, 740 MPa, 19.2%, and 592 MPa, 720 MPa, 17.3% respectively; and in steel B are 535 MPa, 645 MPa, 23.4% in surface region and 485 MPa, 608 MPa, 16.2% in the mid-thickness region. In both steels, the strength decreases with the increase in distance to the surface.

In steel A, the impact energy at −20 °C and −40 °C is 88.9 J and 68.9 J in the surface, while in the mid-thickness, the impact energy at −20 °C and −40 °C is 75.3 J and 55.8 J, respectively. In steel B, the impact energy at −20 °C and −40 °C is 55.6 J and 30.6 J in the surface, while in the mid-thickness, the impact energy at −20 °C and −40 °C is 45.2 J and 28.2 J, respectively. The impact energy decreases with the decrease in tested temperature and is higher in the surface compared with that in the mid-thickness region.

In order to analyze the fracture mechanism, the tensile fracture and impact fracture have been observed in both steels. Figure 6 shows the SEM fractography of tensile fractures in the surface region of steel A and B. The failure mode in both steels exhibit a “cup-cone” type of fracture, and three zones (fibrous, radial, shear lip) can be observed, indicating the fracture mode is ductile fracture. Compared to that in steel B, the fibrous zone is relatively larger, as indexed by the dashed ellipses in Figure 6a,e. The higher magnification of the fibrous zone is presented in Figure 6b,f. Extensive dimples can be observed, accompanied by some micro-pores and macro-cracks. In this region, the deformation is uniform, corresponding to the uniaxial stress state condition. Figure 6c,g present the higher magnification of the radial region. The cleavage facets have been observed, and the crack propagation direction can be confirmed. In this period, the cracks lose their stability and propagate quickly. In most parts of the shear lip zones, the angle between the shear lip zone and axial direction is 45°. This characteristic is indicative of combined normal and shear separation. Part of the shear-lip rim in Figure 6a,e is larger than 45°, suggesting that crack deviation is random in different directions [14]. The enlargement view of the shear lip zone in both steels is presented in Figure 6d,h. The dimples are flat and stretched, and the dimples in steel A are relatively shallow compared with that in steel B. The particles located in the dimples can be observed, and the composition of the particles is TiC, as presented in the inset of Figure 6h.

The typical deflection–load curve of the Charpy impact specimen usually contains five stages [14,25]. The total absorbed energy includes *E*_1_, *E*_2_, and *E*_3_, and refer to crack initiation energy, table crack propagation energy, and unstable fracture energy, respectively, as shown in Figure 7a. Figure 7b,c shows the deflection–load curve of the Charpy impact specimen obtained from the surface region of steel A and B tested at −20 ℃. The total impact absorbed energy in steel A is 88.9 J. The crack initiation energy and crack propagation energy are 37.6 J and 51.3 J, respectively (Figure 7b), while in steel B, the total impact absorbed energy is 55.6 J, and the crack initiation absorbed energy and crack propagation energy are 28.9 J and 26.7 J (Figure 7c). The main difference in the total absorbed energy should be attributed to the crack propagation energy (*E*_2_ and *E*_3_). The relatively high crack propagation energy can be attributed to the fine bainite structure and the high-volume fraction of grain boundary in the surface of steel A, comparing that in steel B.

Figure 8 shows SEM fractography of the tensile test specimens obtained from the surface region of steel A and B at −20 ℃. The macrograph fracture morphology of steel A and B is shown in Figure 8a,d. Three regions can be observed, including fibrous, radial, shear lip zones ignoring the V-notch zone, and the volume fraction of shear lip zone in steel A is higher compared to that in steel B. In steel A, tearing has been observed and in steel B, the river pattern can be observed, which indicates that the impact toughness of steel A is higher compared with that in steel B. Figure 8b,e shows the fractography of the fibrous zone in both steel, dimples, and particles can be observed. Figure 8c,f shows the fractography of the radial zone, referring to the crack propagation process. In this zone, the cracks propagate quickly, and the shear lip zone is produced after that the crack propagation is inhibited by plastic deformation. In general, the fibrous zone and shear lip zone can absorb higher impact energy compared to the radial zone. Therefore, the total absorbed energy in steel A should be higher compared with that in steel B for the higher volume fraction of fibrous and shear lip zones.

### 3.3. Precipitation Behavior

Figure 9 shows the TEM images of precipitates formed in the surface and mid-thickness steel A. Figure 9a shows the bright-field (BF) image of precipitates in PF in the surface. A large number of randomly distributed precipitates have been observed, and the average precipitate size is ~3.5 nm. The SAED pattern illustrated in Figure 9b shows ferrite matrix with [110] direction and precipitates with [112] direction. The result indicates that the precipitates have a NaCl crystal structure, and the lattice parameter of these precipitates is roughly 0.432 nm, which is close to the lattice parameter of TiC. It can also be concluded from the SAED pattern that these precipitates obey [112]_MC_//[110]_α_ and ${\left(\bar{1}\bar{1}2\right)}_{\mathrm{MC}}$//${\left(\bar{1}12\right)}_{\alpha }$ orientation relationship with the ferrite matrix, which is different from the previously reported ones, such as BN and NW orientation relationship. Figure 8c illustrates the TEM image of precipitates formed in the mid-thickness of steel A plate. There are two size ranges of precipitates, and the large precipitates in the size range of 15–20 nm were assumed to be strain-induced during hot-rolling, while the tiny precipitates in the size range of 3–8 nm, were assumed to form in the cooling process. The precipitates are relatively larger compared with that in the surface region.

In order to clearly illustrate the new orientation relationship observed in Figure 9. The structure of TiC and ferrite phases were put into the Crystal Maker software for simulation purposes, as shown in Figure 10. Figure 10a,c presents the simulated TiC and ferrite unit cells with the indexed plane of ${\left(\bar{1}\bar{1}1\right)}_{\mathrm{Tic}}$ and ${\left(\bar{1}\bar{1}2\right)}_{\mathrm{ferrite}}$. Figure 10b,d shows the atom distribution of TiC and ferrite in the zone axis of ${\left[112\right]}_{\mathrm{Tic}}$ and ${\left[110\right]}_{\mathrm{ferrite}\;}$, together with the indexing of ${\left(\bar{1}\bar{1}1\right)}_{\mathrm{Tic}}$ and ${\left(\bar{1}12\right)}_{\mathrm{ferrite}}$ planes. The simulation results reveal the orientation relationship intuitively.

In order to analyze the difference between those two kinds of precipitates, the HRTEM technique was used to analyze the orientation relationship between ferrite and precipitates. Figure 11a shows the HRTEM image of the relatively large precipitate in Figure 9c. The crystal structure and the orientation relationship were identified by a fast Fourier transformed (FFT) diffractogram, as illustrated in Figure 11b. It can be deduced that the precipitate obeys the Kurdjumov–Sachs (KS) orientation relationship with ferrite matrix, that is ${\left[111\right]}_{\mathrm{ferrite}}{//[110]}_{\mathrm{TiC}}$ and ${\left(\bar{1}01\right)}_{\mathrm{ferrite}}{//(111)}_{\mathrm{TiC}}$ [26,27]. To our knowledge, austenite strain-induced precipitates adopt cube-on-cube orientation relationship with austenite, that is ${\left[001\right]}_{\mathrm{ferrite}}{//[001]}_{\mathrm{TiC}}$ and ${\left(001\right)}_{\mathrm{ferrite}}{//(001)}_{\mathrm{TiC}}$, when austenite is transformed into ferrite, the KS orientation relationship obeyed by austenite and ferrite will be inherited by the precipitate and ferrite [28,29]. Figure 11c,d illustrate the IFFT images of the precipitates and ferrite matrix, and the lattice parameter can be measured to be 0.286 nm and 0.432 nm.

Figure 12 shows the TEM image of precipitates in the surface and mid-thickness of steel B. In the surface region of steel B, the precipitates are in the size range of 5–20 nm, while in the mid-thickness region, the precipitates are in the size range of 10–30 nm, as shown in Figure 12a,b. The EDS result indicates that the precipitate is TiC, as shown in Figure 12c.

### 3.4. Strengthening and Toughing Mechanism

For low-carbon steel, the yield strength equals to the sum of solid strengthening, grain refinement strengthening, and precipitation strengthening, and is given by the following equation [21,26].
(1)$$\begin{array}{l} \sigma_{y}=\sigma_{SG}+\sigma_{SS}+\sigma_{SP}\\ =600D^{-1/2}+\{46[C]+37[\mathrm{Mn}]+83[\mathrm{Si}]+59[\mathrm{Al}]+2918[N]+80.5[\mathrm{Ti}]\}+\sigma_{SP} \end{array}$$
where, *σ*_Y_, *σ*_SG_, *σ*_SS_, and *σ*_SP_ represent the yield strength, grain refinement strengthening, solid solution strengthening, and precipitation strengthening in MPa, respectively. *D* represents the average grain size. From Equation (1), the solid solution strengthening has been determined to be ~80 MPa and it is concluded that the main factors affecting yield strength are grain refinement strengthening and precipitation strengthening.

The precipitation hardening contribution to yield strength was estimated by Ashby-Orowan [26]:(2)$$\Delta \sigma =10.8\frac{\sqrt{f}}{d}\ln(1630d)$$
where, *f* is the volume fraction of precipitates and *d* is the average radius of the precipitates. The precipitation strengthening in the surface and mid-thickness region of steel A was calculated to be 320.4 MPa and 260.9 MPa, while in steel B, the precipitation strengthening is 230.6 MPa and 218 MPa.

In order to calculate the grain refinement strengthening, the equivalent grain size should be measured. Figure 13 illustrates the EBSD analysis of steel A, including grain size evaluation and grain boundary analysis, which determined the deflection of crack propagation, that is impact toughness. The grain size in the surface and mid-thickness region are 6.5 μm and 7.3 μm, as shown in Figure 13a,c,e,g). Figure 13b,f shows the image quality map, the black lines mean high misorientation grain boundaries (HAGB, ≥15°), and red lines mean low misorientations grain boundaries (LAGB, 2–15°). The grain boundaries of AF and PF, packets, and blocks of LB exhibit HAGBs, while the laths inside the austenitic grains show LAGBs. The HAGB can effectively deflect the propagation of cleavage microcracks, whereas the LAGB have less ability to deflect the crack [26,30]. The fraction of HAGB in surface and mid-thickness are 57% and 39%.

Based on Equation (1), the grain refinement strengthening has been calculated to be 235.3 and 222.1 MPa in the surface and mid-thickness region of steel A, while in steel B, the grain refinement strengthening is 165.2 and 139.6 MPa. So, the main difference in yield strength for steel A and B should be attributed to grain refinement strengthening and precipitation strengthening. In addition, it is visible that UFC can unify the grain size and precipitate size, which guarantees the homogeneity of yield strength. In addition, the excellent impact toughness in steel A can be attributed to the high-volume fraction of LB and AF and the small polygonal ferrite grain, which have a strong ability to deflect the crack propagation.

Above all, the mechanical properties in steel A are more homogeneous compared with that in steel B, which can be attributed to the relatively uniform cooling rate in the newly developed UFC system.

## 4. Conclusions

(1)An excellent heavy steel plate has been obtained with a yield strength, tensile strength, and impact toughness of −20 ℃ of 592 MPa, 720 MPa, and 75 J, which is mainly devoted to grain refinement strengthening and precipitation strengthening.(2)The microstructure and mechanical properties of the heavy steel plate produced by UFC are more uniform compared with that by traditional accelerated cooling, for the cooling rate of UFC is uniform along the longitudinal direction of the thick plate.(3)The crystallographic characteristic of precipitates has been analyzed. The size of austenite strain-induced TiC is relatively large, and obeys the KS orientation relationship with the ferrite matrix, while for TiC formed in the supersaturated ferrite, the size is less than 5 nm, and obeys [112]_MC_//[110]_α_ and ${\left(\bar{1}\bar{1}2\right)}_{\mathrm{MC}}$//${\left(\bar{1}12\right)}_{\alpha }$ orientation relationship with the ferrite matrix.

## Figures and Tables

**Figure 1 materials-15-01385-f001:**
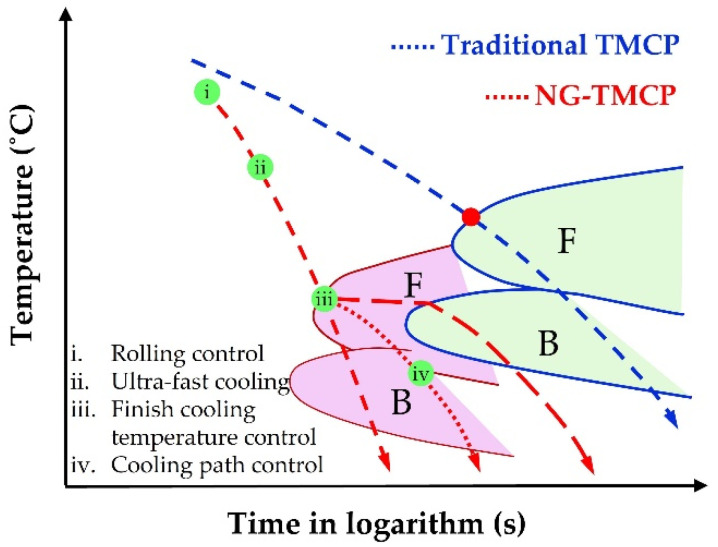
Schematic diagram showing the difference between NG-TMCP and traditional TMCP. F stands for ferrite, and B stands for bainite in this figure.

**Figure 2 materials-15-01385-f002:**
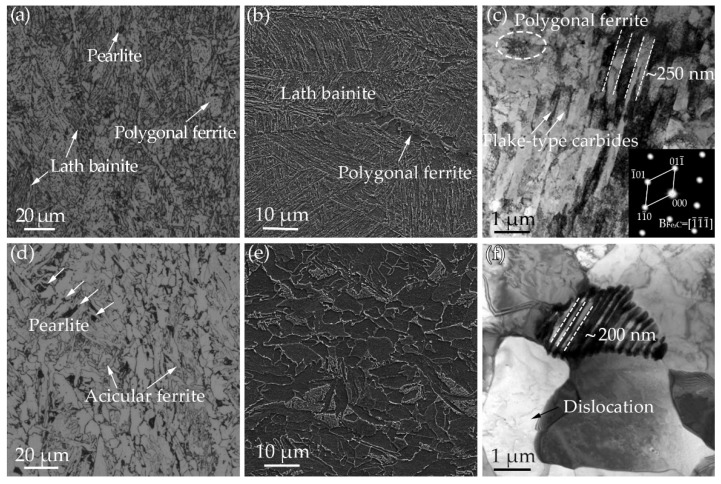
(**a**,**d**) OM images; (**b**,**e**) SEM images and (**c**,**f**) TEM images of the specimens obtained from surface and mid-thickness of in steel A plate.

**Figure 3 materials-15-01385-f003:**
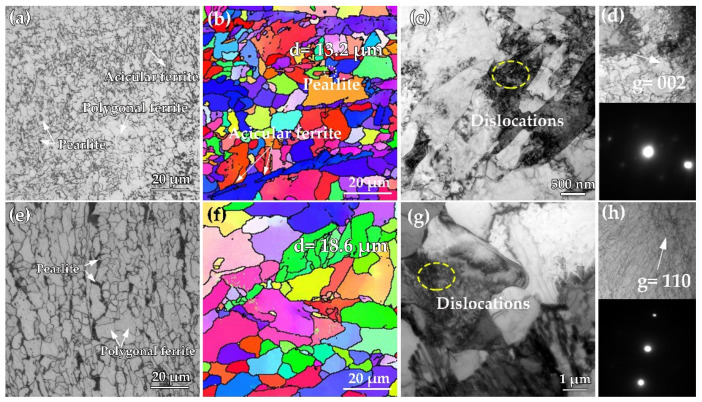
Microstructure analysis of the specimens obtained from the surface and mid-thickness region of steel B. (**a**,**e**) OM images; (**b**,**f**) EBSD micrographs; (**c**,**g**) TEM images; (**d**,**h**) the enlargement view of yellow circle in (**c**,**g**) showing the dislocations observed by two-beam diffraction methods in different vectors.

**Figure 4 materials-15-01385-f004:**
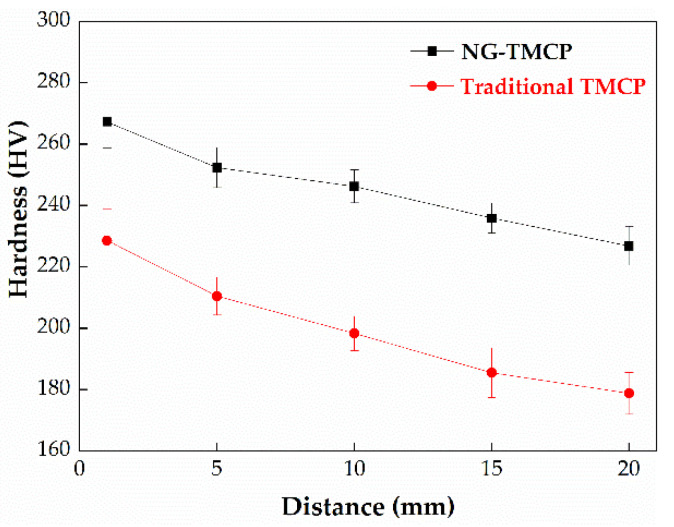
Vickers hardness evolution from the surface to the center part.

**Figure 5 materials-15-01385-f005:**
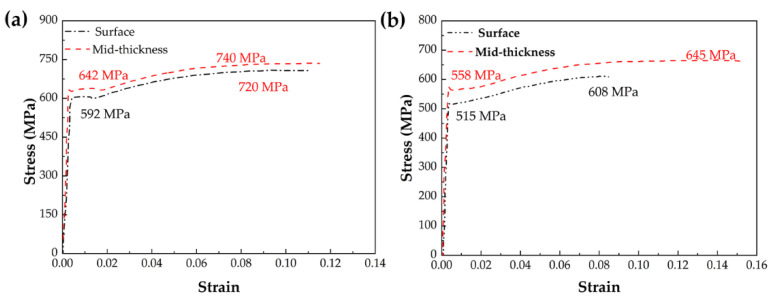
Tensile stress–strain curves of specimens obtained from surface and mid-thickness of steels A and B. (**a**) Tensile stress–strain curves of steel A; (**b**) Tensile stress–strain curves of steel B.

**Figure 6 materials-15-01385-f006:**
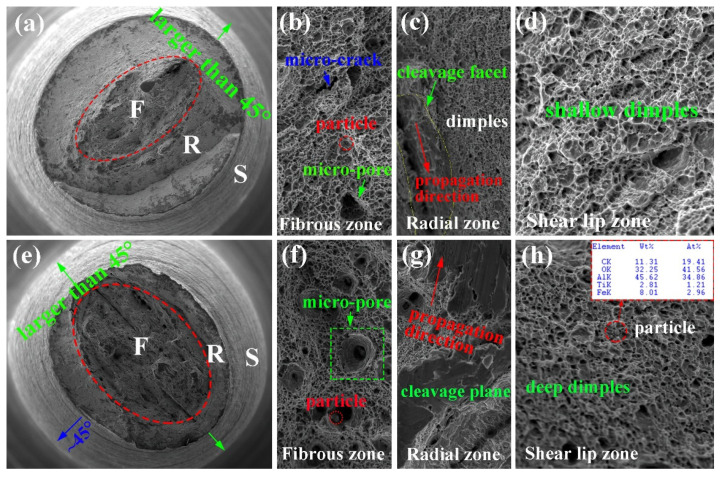
SEM micrographs of the tensile fracture of the specimen in the surface of steel A and B. (**a**,**e**) low magnification of the fracture, F stands for fibrous, R stands for radial, and S stands for shear lip; (**b**,**f**) high magnification of fibrous zone; (**c**,**g**) high magnification of radial zone; (**d**,**h**) high magnification of shear lip zone.

**Figure 7 materials-15-01385-f007:**
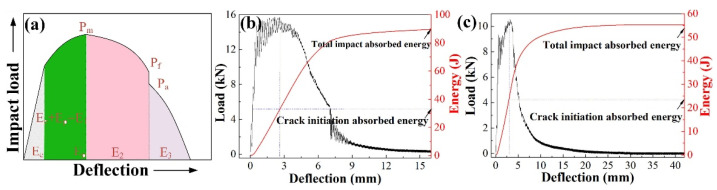
Schematic diagram (**a**) and deflection–load curves of the impact test specimen in steel A and B (**b**,**c**).

**Figure 8 materials-15-01385-f008:**
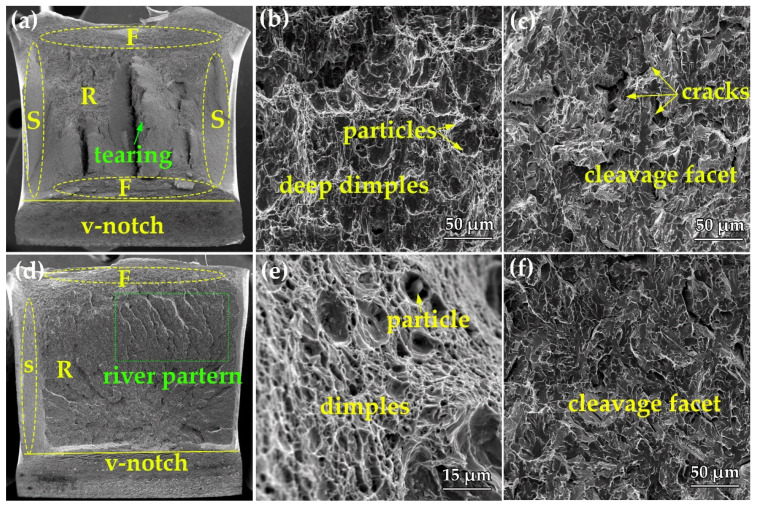
The impact fractography of the specimens obtained from the surface of steel A and B. (**a**,**d**) macrographs; (**b**,**e**) the fractography of shear lip zone; (**c**,**f**) the fractography of radial zone.

**Figure 9 materials-15-01385-f009:**
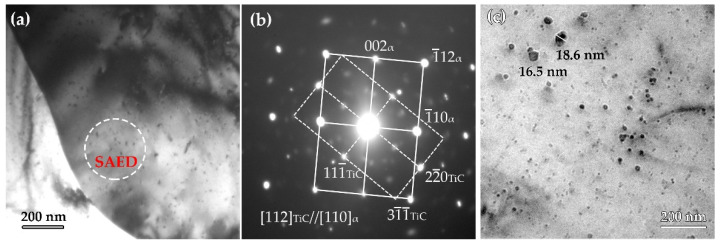
TEM images of precipitates formed in steel A. (**a**) precipitates in surface region; (**b**) SAED pattern, (**c**) precipitates in mid-thickness region.

**Figure 10 materials-15-01385-f010:**
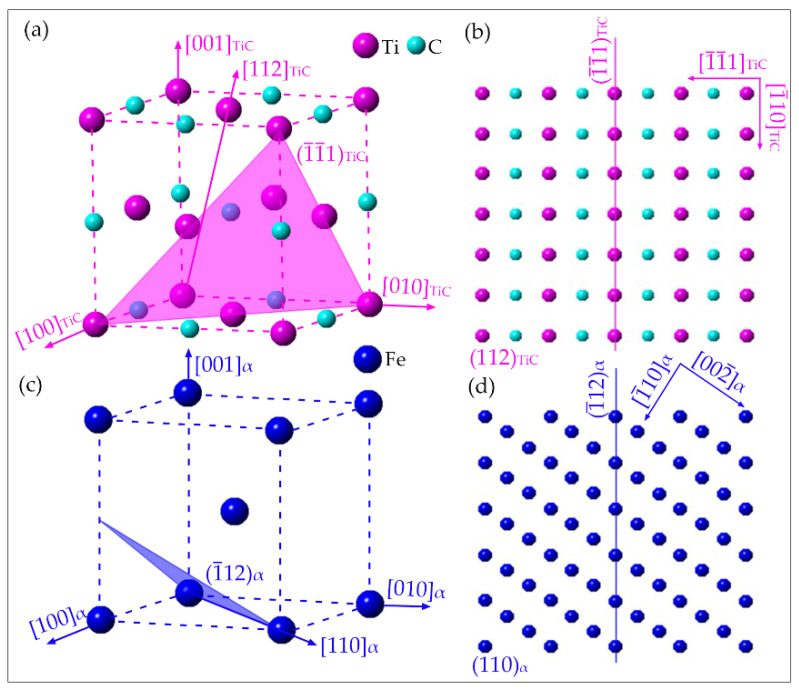
Simulation result of the new orientation relationship by Crystal Maker software. (**a**) unit cell of TiC with the indexing of ${\left(\bar{1}\bar{1}1\right)}_{\mathrm{Tic}}$ plane; (**b**) unit cell of ferrite with the indexing of ${\left(\bar{1}12\right)}_{\mathrm{ferrite}}$ plane (**c**,**d**) atom distribution of TiC and ferrite in the zone axis of ${\left[112\right]}_{\mathrm{Tic}}$ and ${\left[110\right]}_{\mathrm{ferrite}}$.

**Figure 11 materials-15-01385-f011:**
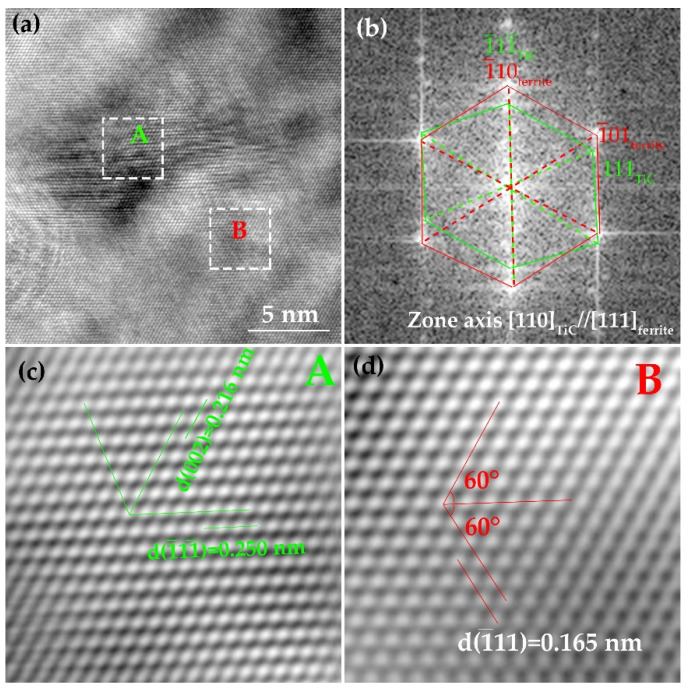
Crystallographic characterization of deformation-induced TiC formed in the mid-thickness of the plate in steel A. (**a**) HRTEM image; (**b**) FFT diffractogram; (**c**,**d**) IFFT of precipitate and matrix.

**Figure 12 materials-15-01385-f012:**
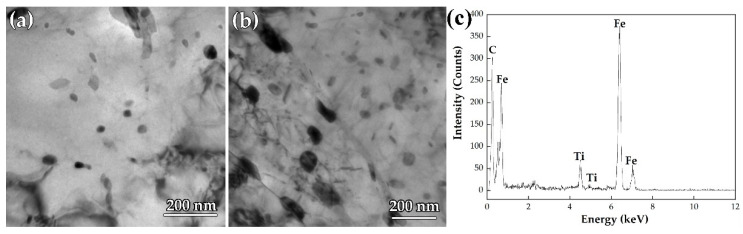
(**a**,**b**) TEM images of the precipitates obtained from surface and mid-thickness and (**c**) the corresponding EDS of steel B plate.

**Figure 13 materials-15-01385-f013:**
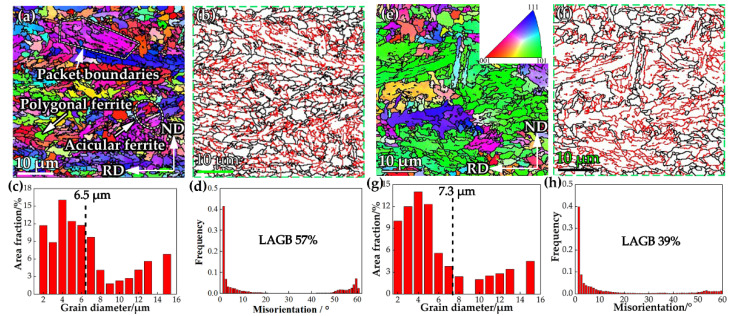
EBSD analysis of the specimens obtained from surface and mid-thickness region in steel A plate. (**a**,**e**) orientation image map, (**b**,**f**) image quality map with grain boundary misorientation, (**c**,**g**) grain diameter distribution, and (**d**,**h**) misorientation distribution.

**Table 1 materials-15-01385-t001:** The chemical composition of the experimental steel in this study (wt.%).

C	Mn	Si	Al	Ti	P	S	N	O
0.15	0.98	0.28	0.02	0.08	0.015	0.005	27 ppm	48 ppm

**Table 2 materials-15-01385-t002:** Thermal mechanical processing parameters of the tested steels.

No.	RTG ^1^ (°C)		NRTG ^2^ (°C)		Cooling Rate, °C/s	FCT ^3^, °C	Type of Cooling
	Start	Finish	Start	Finish			
A	1150	1091	880	875	60 (45)	590 (621)	20 min, air cooling
B	1150	1096	889	864	10 (5.8)	598 (678)	20 min, air cooling

^1^ RTG: recrystallization temperature region. ^2^ NRTG: non-recrystallization temperature region. ^3^ FCT: finish cooling temperature.

**Table 3 materials-15-01385-t003:** Mechanical properties obtained from the surface and mid-thickness of steel A and B plates.

Steel		Yield Strength, MPa	Tensile Strength, MPa	Elongation, %	Yield Ratio	Impact Toughness, J	
						−20 °C	−40 °C
A	S ^1^	642 (5.2)	740 (6.1)	19.2 (1.3)	0.87 (0.2)	88.9 (3.6)	68.9 (5.4)
	M ^2^	592 (5.7)	720 (5.9)	17.3 (1.8)	0.82 (0.2)	75.3 (2.8)	55.8 (5.1)
B	S	558 (2.8)	645 (2.1)	23.4 (0.9)	0.86 (0.1)	55.6 (5.6)	30.6 (6.2)
	M	515 (4.3)	608 (4.9)	16.2 (1.2)	0.85 (0.1)	45.2 (4.2)	28.2 (8.1)

^1^ S represents the surface of the steel plate. ^2^ M represents the center thickness of the steel plate.

## Data Availability

Not applicable.

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
