# Peer review of "The Effect of Cooling Rate on the Microstructure Evolution and Mechanical Properties of Ti-Microalloyed Steel Plates"

_materials, 2022, doi:10.3390/ma15041385_

Round 1

Reviewer 1 Report

The submitted manuscript entitled ‘The determine role of cooling rate on the microstructure evolution and mechanical properties in the thick plates of Ti-microalloyed steel’ is dealing with the intensive microstructural (OM, SEM, EBSD, TEM) and mechanical (hardness, tensile properties) investigations of a relatively low carbon content, Mn, Si, Al, Ti alloyed, presumable case hardening steel (maybe for nitriding). The main concern of this Reviewer is in the lack of scientific contributions. The Conclusions consist of observations only. Moreover, the manuscript is relatively poorly written (in English).

  1. English is not the native language of this Reviewer, but a proof-reading is strongly recommended. Even the Title itself is confusing.
  2. Please remove the commercial e-mail address from the corresponding Author list.
  3. ‘The chemical composition of the investigating steel was 0.15%C-0.98%Mn-0.28%Si- 66
  4. 02%Al-0.08%Ti-Fe (wt. %).’ – how was this composition determined?
  5. Line 67: ‘cast’ instead of ‘casted’.
  6. How was the ‘ultra-fast cooling’ performed? Please add details (cooling medium, temperature, etc.).
  7. Fig 1: please add axis labels.
  8. Please add grinding details (before etching).
  9. The orientation of the tensile samples is unclear. Are they in plane or perpendicular to the plane of the plate both can be perpendicular to the rolling direction)?
  10. Line 105: please use superscript in the dimension.
  11. Line 184: ‘precipitations’ instead of ‘precipitates’.
  12. Fig 10: the text mentions distance from the middle plane, while the graph shows the distance from the surface. It is confusing, please unify.
  13. Table 2: caption is missing. Please add standard deviations for the mechanical properties.

Reviewer 2 Report

Comments and specific questions requiring clarification:

  1. Introduction in my opinion is a little to short– Please add some more information (reference) what is done (explained) up to this time in technical literature in this technical problem. Maybe You can for example show some diagrams of influence of different TMCP method on microstructure and mechanical properties – If You find it in technical literature.
  2. Punkt 2.1 – I recommend to show chemical composition in the table – it will be more radable.
  3. If it is possible please add some information about cooling system used for ultra fast cooling (cooling medium etc)
  4. Why the samples for investigations of mechanical properties was made as perpendicular to the rolling direction while the samples for impact toughness were made as along to the rolling direction?
  5. Fig 3c-3h – make the axis description larger please
  6. Fig 14 – make the figure (especially axis description) larger please and improve quality of the picture
  7. If it is possible I recommend to don’t divide the words at the end of line – so if it is possible please correct it in the whole paper

Reviewer 3 Report

Major remarks:

1/ Title. „The determine role …” It seems incorrect. Please reconsider this.

2/ „Therefore, it is essential to propose an effective way to strengthen 31 the microalloyed steels and make the microalloyed steels develop continuously and pos- 32 itively [1-7].” Individual achievements and results of the works should be indicated. I mean …[1] …[2]…etc instead of blocky citation without the insight into the real literature review.

3/ More details on the microalloying effects in thermomechanically processed plates should be provided in Introduction. For example, please see: doi: 10.3390/met8050304, doi: 10.3390/met8121028

4/ Figure 1. A lack of axis captions

5/ How were RTG: recrystallization temperature region and NRTG: non-recrystallization temperature region determined ?

6/ Hardness load must be provided in N, as SI unit

7/ Figure 3. The figure captions are not visible

8/ Some statistics for the data in Table 2 should be provided; standard deviation, etc.

9/ In the present form this is a nice (especially TEM work) technical report only. The scientific discussion must be provided with emphasis on available literature data.

Round 2

Reviewer 1 Report

The submitted manuscript is dealing with the intensive microstructural (OM, SEM, EBSD, TEM) and mechanical (hardness, tensile properties) investigations of a relatively low carbon content, Mn, Si, Al, Ti alloyed, presumable case hardening steel (maybe for nitriding). The main concern of this Reviewer is still in the lack of scientific contributions. The Conclusions consist of observations only. This fact has not been explained in the responses.

Moreover points 6 and 7 of the original review was not properly answered.

  1. Fig 1: please add axis labels. The time axis should be on a logarithmic scale.
  2. Please add grinding details (before etching).

Reviewer 3 Report

The authors improved a little bit the content of the manuscript. However, some further revision is obligatory before possible acceptation:

1/ You added some new positions in Introduction. However, they are not cited in the text. Hence, there is a wrong nemeration of references.

2/ English must be improved in the whole manuscript.

3/ Still, the real scientific discussion is lacking. The data are interesting. However, they must be discussed in relation to what was done so far in the literature. Without it, the manuscript is a technical report only. 
